# Application of Factorisation Methods to Analysis of Elemental Distribution Maps Acquired with a Full-Field XRF Imaging Spectrometer

**DOI:** 10.3390/s21237965

**Published:** 2021-11-29

**Authors:** Bartłomiej Łach, Tomasz Fiutowski, Stefan Koperny, Paulina Krupska-Wolas, Marek Lankosz, Agata Mendys-Frodyma, Bartosz Mindur, Krzysztof Świentek, Piotr Wiącek, Paweł M. Wróbel, Władysław Dąbrowski

**Affiliations:** 1Faculty of Physics and Applied Computer Science, AGH University of Science and Technology, al. Mickiewicza 30, 30-059 Kraków, Poland; bartlomiej.lach@fis.agh.edu.pl (B.Ł.); tomasz.fiutowski@agh.edu.pl (T.F.); stefan.koperny@fis.agh.edu.pl (S.K.); mlankosz@agh.edu.pl (M.L.); bartosz.mindur@agh.edu.pl (B.M.); swientek@agh.edu.pl (K.Ś.); wiacek@agh.edu.pl (P.W.); pwrobel@agh.edu.pl (P.M.W.); 2Laboratory of Analysis and Non-Destructive Investigation of Heritage Objects, The National Museum in Krakow, al. 3 Maja 1, 30-062 Kraków, Poland; pkrupska@mnk.pl (P.K.-W.); amendys@gmail.com (A.M.-F.)

**Keywords:** XRF spectral imaging, image processing, micropattern gaseous detectors

## Abstract

The goal of the work was to investigate the possible application of factor analysis methods for processing X-ray Fluorescence (XRF) data acquired with a full-field XRF spectrometer employing a position-sensitive and energy-dispersive Gas Electron Multiplier (GEM) detector, which provides only limited energy resolution at a level of 18% Full Width at Half Maximum (FWHM) at 5.9 keV. In this article, we present the design and performance of the full-field imaging spectrometer and the results of case studies performed using the developed instrument. The XRF imaging data collected for two historical paintings are presented along with the procedures applied to data calibration and analysis. The maps of elemental distributions were built using three different analysis methods: Region of Interest (ROI), Non-Negative Matrix Factorisation (NMF), and Principal Component Analysis (PCA). The results obtained for these paintings show that the factor analysis methods NMF and PCA provide significant enhancement of selectivity of the elemental analysis in case of limited energy resolution of the spectrometer.

## 1. Introduction

The X-ray Fluorescence (XRF) spectroscopy is a well-established technique used for the investigation of the elemental composition of various materials including geological and biological samples as well as various types of artworks. As long as the intensity of X-rays is kept within certain limits, this can be considered as a noninvasive technique, and, as such, it is suitable for the investigation of surface layers of valuable art objects. The technique is widely used for the investigation of paintings [1,2,3,4,5,6].

The macro-XRF technique utilises a focused X-ray microbeam for excitation of fluorescence radiation, a mechanical scanning system, and a high energy resolution X-ray detector, typically a silicon drift detector. Thus, the spatial resolution in this method is determined primarily by the spot size of the exciting beam. The technique is very suitable for detailed investigation of small area samples like fragments of large paintings. When applied to the investigation of large area painting, it becomes quite time consuming, although much progress has been made recently in various aspects of this method, including X-ray tubes, radiation detectors, mechanical scanning systems, and software. There is a commercially available instrument from Bruker Nano GmbH known as M6-Jetstream [7], which is used nowadays for investigation of flat cultural heritage objects, like paintings, icons, manuscripts, and stained glass [6,8,9,10,11,12]. It is worth noting that the instrument has been developed addressing specific requirements of cultural heritage studies. It uses a state-of-the-art silicon drift X-ray detector providing the energy resolution at the level of about 2.5% Full Width at Half Maximum (FWHM) at 5.9 keV. In parallel, custom-designed instruments are being developed, e.g., [13,14,15,16,17]. Applications of the macro-XRF technique are limited to flat objects like paintings. For nonflat objects, the spatial resolution gets degraded if the investigated surface is not precisely in the focal plane of the microbeam [18,19].

An alternative technique called full-field imaging has been proposed recently, and several projects are under development [20,21,22]. The full-field imaging technique requires a position-sensitive and energy-dispersive X-ray detector. The investigated object is illuminated by a broad X-ray beam, and the fluorescence radiation is projected onto a 2-D position-sensitive detector through an optical system like, for example, a multihole collimator or a pin-hole camera, as shown in Figure 1. There is no obvious choice for the type of detector for such applications as typically the energy resolution of position-sensitive detectors is compromised by physical effects, like charge division [23], and by technical constraints. Different types of detectors have been tried by other groups for such applications: a detector based on a Charge Coupled Device (CCD) used in a special spectroscopic mode [21,24,25,26], a pixel detector [27,28], a gaseous Micro Hole Strip Plate (MHSP) detector [29], and a gaseous detector based on the technology of Thick Gas Electron Multiplier (THGEM) [22], called THCOBRA. Application of the Gas Electron Multiplier (GEM) detector with the resistive divider readout has also been tried [30].

We developed a system employing a gaseous detector based on the standard (thin foil) GEM technology. The proof of principle of full-field imaging has been demonstrated using a three-stage GEM chamber with a sensitive area of 10 × 10 cm${}^{2}$ and a 2-D strip-like readout working with an Ar/CO${}_{2}$ (70/30) gas mixture [20]. Following this step, further improvements in the GEM detector by introducing Cr-clad foils have been worked out [31]. Despite improvements in the GEM detector and the readout system, the energy resolution of GEM detectors working in the proportional counter mode is moderate and limited to about 18% FWHM at 5.9 keV due to fluctuation of the charge generated in the detector. Therefore, the interpretation of the XRF spectra measured with the GEM detectors is not as straightforward as in the case of high energy resolution silicon detectors used typically in macro-XRF instruments. In addition, the measured XRF spectra include escape peaks specific for given gas mixtures, which further complicate the interpretation of the spectra with multiple characteristic energy lines.

Using the developed instrument, we investigated about 20 different paintings so far. To obtain data of the best possible quality, the specific calibration and measurement procedures discussed later in this article were worked out. The analysis and interpretation of the data is as equally important and challenging as the optimisation of the hardware and measurement procedures. In this article, we discuss the application of factor analysis methods, Principal Component Analysis (PCA) and Non-Negative Matrix Factorisation (NMF), for analysis of the data delivered by our instrument. Both methods use an orthogonal transformation to convert a set of possibly correlated data into a set of linearly uncorrelated data. They are widely employed in various fields of science from biology, neuroscience, medicine, astronomy, and X-ray computed tomography, using various types of physical data. In particular, applications of these two methods to analysis of XRF-imaging data for historical paintings obtained with different detection techniques are reported in the literature [32,33,34]. The PCA is sometimes criticised by the lack of direct interpretability of the basis vectors as they can comprise negative values, while the input data comprise only non-negative values. The NMF method constrains the results of factorisation to non-negative values, and the interpretation of the resulting basis vectors can be more intuitive. It is worth noting that the interpretation of the measurement and decomposition data is specific for each particular object and requires some level of a priori knowledge about it. In this work, we present the analysis for two selected objects and discuss the results obtained by the two factorisation methods.

## 2. Materials and Methods

### 2.1. Overview of the Full-Field XRF Imaging Spectrometer

In our work, we used a custom-developed full-field XRF imaging spectrometer. The conceptual design of the spectrometer is described elsewhere [35]. The key components of the instrument are: two X-ray tubes (Varian VF-50J 50 W with molybdenum anode), a custom-designed pin-hole camera with the possibility of selecting different hole diameters from 1 mm to 2 mm, GEM position-sensitive detector, and a custom-designed readout system of the detector including Application Specific Integrated Circuit (ASICs). Two X-ray tubes were used to illuminate the area of 10 × 10 cm${}^{2}$ as uniformly as possible. The details of the X-ray optics system are described in [35]. In the basic configuration, the system was set up with the optical magnification of 1, the pin-hole diameter of 2 mm, and a 65 mm distance between the pin-hole and the investigated object. Thus, for the detector with the active area of 10 × 10 cm${}^{2}$, we obtained an image from the area of the same size. The spatial resolution was determined by the pin-hole diameter. According to the model presented in [36], for the pin-hole diameter of 2 mm, the spatial resolution due to the pin-hole was 1.7 mm${}_{rms}$ in the central part of the detector, while the intrinsic spatial resolution of the detector with a 0.8 mm readout pitch was 0.23 mm${}_{rms}$ (pitch/$\sqrt{12}$). The spectrometer head was mounted on the arm of an industrial robot, which allows one to position the head precisely in front of an investigated object and move it to take multiple images with steps of 10 cm. A schematic view of the system is shown in Figure 2a, and a photo of the measurement head mounted on the robot arm is shown in Figure 2b.

### 2.2. GEM Detector

Development of detectors based on the GEM technology is primarily driven by their applications in high energy physics experiments where they are used as position-sensitive detectors in particle tracking systems. In such applications, the primary goal is to obtain the best possible spatial resolution for relativistic charge particles. To achieve good detection efficiency and a low noise count rate, a good signal-to-noise ratio is required. Furthermore, given that the collected charge is spread over several readout electrodes, strips, or pixels, one can obtain a significant improvement in the spatial resolution by utilising the amplitudes of signals recorded at individual electrodes within the cluster. The centre of gravity of such a cluster points to the most probable position of particle interaction with the detector’s active volume.

Detectors based on the GEM technology have been proven to be also suitable for simultaneous spectral and position-sensitive measurements of low-energy X-rays, up to about 20 keV. For higher energies, the detection efficiency of such detectors would be too low for various practical applications, although one can extend the usable energy range by using gas mixtures based on heavier gases like krypton or xenon instead of commonly used cheap argon. One can also increase the detection efficiency by operating the detectors at a higher gas pressure, above the atmospheric pressure, but this would require a more advanced construction of the entrance window instead of the commonly used kapton foil.

The energy resolution of such detectors is determined mainly by fluctuations in the charge produced in the impact ionisation processes, and they cannot compete with semiconductor detectors. However, the GEM detectors have other advantages that make them attractive for some applications in the field of X-ray imaging. First of all, detectors with large areas can be manufactured easily at low cost. Furthermore, they do not require cooling and can be operated at room temperature. The geometry and technology of the readout electrodes are decoupled from the GEM technology, and various types of readout structures can be easily implemented according to a given requirement. Of course, the readout electronics have to be based on ASICs like for any other high-granularity position-sensitive detector.

When aiming at the best possible energy resolution, approaching the physical limits due to fluctuations of the generated charge, one has to pay attention to other effects, namely, variation in the gas amplification factor across the detector and dependence of the gas amplification factor on the radiation intensity. In the literature, one can find very different numbers on the energy resolution of GEM detectors from 18% to 27% FWHM for X-ray energy of 5.9 keV [37,38,39,40]. The energy at the level of 18% FWHM is usually measured using a focused X-ray beam so that the gas amplification factor is constant over a small examined area of the detector. On the other hand, if the detector is illuminated with a broad beam, the spatial variations of the gas amplification factor contribute to the spread of the measured signal, which results in further worsening the energy resolution.

The map of the gas amplification factor for the detector used in the spectrometer, extracted from the measurements of the characteristic radiation of copper projected on the entire detector area, is shown in Figure 3a. The gas gain map after off-line correction is shown in Figure 3b. It is worth noting that the spread of the gas gain was suppressed from the initial ±30% min–max to the ±1‰ min–max after correction. Correction of the gas gain spread across the detector area is a key step towards application of the GEM detector to spectral imaging of X-rays. After off-line correction of the gas amplification factor, the X-ray spectrum summed over all pixels shows essentially the same energy resolution as one could obtain from the measurement with a focused beam. An example of the cumulative spectrum of the copper characteristic radiation is shown in Figure 4. The energy resolution was 18% FWHM.

In addition, when aiming at the best possible energy resolution, one also needs to pay attention to variation in the gas amplification factor due to charging-up effects. Although in general the effect is understood as caused by the charge accumulated on the high-resistance kapton foils, which modify the electric field inside the holes, the details depend on many factors. Therefore, the results available in the literature are not always consistent. Most studies suggest that the charging-up saturates and that the gas gain remains stable after initial exposure to radiation on the time scale of hundreds of seconds [41]. However, our observations indicate that, in addition to this global charging-up effect, there is another level of gas gain variation caused by short-term variation in the rate of incident radiation. This observation is consistent with other reported studies [42]. Furthermore, this rate effect is local and contributes to the spatial spread of the gas gain only if the intensity of incident photons varies across the detector area.

Another issue to be addressed is the measurement of X-rays with energies above the copper absorption edge. Since the typical GEM detector is built of copper-clad kapton foils, X-rays of higher energy excite X-ray fluorescence radiation of copper in the active detector volume. This copper fluorescence background limits the detection levels of the elements with characteristic X-ray lines close to the copper K${}_{\alpha }$ line of 8.05 keV and K${}_{\beta }$ line of 8.90 keV. To overcome this serious limitation in the application of GEM detectors to full-field XRF imaging, an additional technology step in production of GEM foils was applied to remove the copper cladding layer from standard GEM foil, leaving only the adhesive chromium layer on the kapton foil and narrow copper stripes. Such a relatively simple modification results in a very significant reduction in the Cu fluorescence background, by a factor of 7 for the Ar/CO${}_{2}$ (70/30) gas mixture [31]. The results of detailed studies of such copperless detectors have been reported elsewhere [43,44].

### 2.3. Readout Electronics and Data Acquisition System

As mentioned above, the energy resolution of large-area GEM detectors is affected strongly by variation in the gas amplification factor across the detector area and in time due to charging-up and count rate effects. To cope with these effects, one needs a versatile readout system capable of measuring the maps of gas gain to be used for correction of the effective signal gain pixel by pixel. In the standard Cartesian detector readout plane, the X- and Y-readout strips are laid out with a pitch of 400 μm; however, the effective readout pitch may be changed by the pitch of the electronic readout channels. The results presented in this article are based on the measurements taken with a readout pitch of 800 μm, i.e., for pairs of neighbouring strips connected to individual channels of the front-end electronics. Such a scheme results in 128 readout channels for each coordinate and 128 × 128 pixels for the entire detector, each of 800 μm ×800 μm.

The key component of our readout system is the low-noise ASIC, called ARTROC, capable of simultaneous measurement of signal amplitude and hit time [45]. The time stamps associated with each hit are used for: (i) merging the signals within the clusters spread over multiple strips and (ii) identification of coincidences of the signal recorded on X- and Y-strips to determine 2-D positions. The photon positions are assigned to the pixels resulting from crossing of the X- and Y-strips with the highest signals within the reconstructed clusters for each coordinate. The ARTROC ASIC comprises derandomising buffers, analogue memory to store signal amplitudes, and digital memory to store encoded time stamps, which are read out through the token-ring based multiplexers. Such a readout scheme results in derandomisation with full zero suppression and allows us to save resources and use only one data acquisition board serving 128 channels for each coordinate.

A simplified block diagram of the readout system is shown in Figure 5. The details of the readout system and event reconstruction software are described elsewhere [46].

### 2.4. Measurement Procedures and Building Data Sets

The ultimate goal of the measurements performed using the developed instrument was to obtain the spatial distribution of elements in the investigated objects. This requires recording and analysing of XRF spectra for individual pixels. However, due to the detector effects related to nonuniformity of the gas amplification factor across the detector, charging-up effects, and variation in the gas amplification factor with the radiation rate, careful measurement procedures should be applied to minimise these effects, and furthermore the recorded raw data needs to be carefully calibrated.

The standard measurement sequence consists of the following steps:Apply the high voltage detector bias for at least 10 h before the planned measurement. This step ensures the stabilisation of the gas amplification factor with respect to the polarisation effects of the kapton inside GEM foils.Measurement of XRF radiation from a dummy copper layer. A single-sided Printed Circuit Board (PCB) with a copper layer of 35 μm thickness was used. The PCB is illuminated for 15 min before starting collecting data, and then the data are recorded during a period of 4 min. There is a twofold aim of this step: (i) stabilisation of the gas amplification factor for a given rate of X-rays and (ii) collecting the data for building the map of the gas amplification factor across the detector.Measurement of the object under investigation. After positioning the tested object, the first frame is illuminated again for 15 min before collecting the data. Because the average intensity of the fluorescence radiation from the object will usually be lower than from the dummy copper foil, it is desirable to let the gas gain to stabilise at different count rates. Of course, we will encounter further small variation of the gas amplification factor when moving from one frame to another one as there are different compositions of elements in different quantities in different regions of the object. Furthermore, since the charging-up effect due to varying X-ray intensity is local, we have to take into account that the gas gain will vary locally across the detector. After the initial stabilisation step, the spectrometer head scans automatically the investigated area frame by frame according to a predefined route, and the apparatus does not require any assistance of the operator.

Processing of the raw collected data is divided into two stages: (i) initial calibration of data and merging data from individual frames and (ii) local calibration of the energy scale to cope with the local charging-up effects. The initial calibration of the data and building of a common dataset for the whole investigated object is performed in three steps:Calculation of the gain map based on data for the dummy copper layer.Correction of the count rates for the vignetting effects introduced by the pin-hole camera. This correction is performed using the same data collected for the dummy copper layer as used for calculation of the gain map.Merging data from all frames into one dataset for the whole investigated area. In this step, the overlapping edges of adjacent frames are removed. Scanning of the spectrometer head across the investigated area is programmed with small overlaps of individual frames.

It should be noted that, at this stage, the data are not corrected for the local count rate effects related to charging-up effects, due to variation in X-ray intensity from different regions of the investigated object. Therefore, another level of energy calibration needs to be performed for much smaller detector regions. This is achieved in an iterative procedure including the following steps:The cumulative energy spectrum is built for the whole investigated area using the data after initial calibration and summing the spectra for individual pixels. Comparing such a spectrum with the spectrum obtained for the dummy copper layer, one can make preliminary assignments of the spectrum peaks to the characteristic energy lines, which are expected in the XRF radiation. At this point, we can utilise other information about the investigated object like, for example, an expected set of pigments associated with the edge of the object. Based on this initial qualitative analysis, we define the list of elements expected to be found in the investigated object.Local energy scale is calibrated for small cells, each one comprising 4×4 pixels. The area of the basic cell used at this stage was selected as a compromise between the accuracy of energy calibration and the statistics. The data from 16 pixels are summed up, and the spectrum for such a cell is created. Since the count statistics within individual cells are rather low, the spectra are smoothed by applying a low-pass filter imported from the Python *statsmodels* library [47]. Then, for such smoothed spectra, the *find_peaks()* function is applied [48], which returns some number of peaks depending on the spectra composition in the given cell. An example of such spectra from four different cells for a particular painting is shown in Figure 6. One can easily notice that because of low statistics, the analysis of such spectra may be non-trivial.A key point in the analysis is the association of the peaks with the specific energy lines. We assume that not all peaks found in the initial analysis of non-corrected data may be present in the data for the given cell. To find the correct or most probable assignment of the peaks, all possible combinations are checked, and the one with the best matching is selected. For example, if the peak finding procedure identifies three peaks in the spectrum and we have six potential candidates, the procedure tries to assign the three peaks to different patterns of the three peaks in the cumulative spectrum. The best matching pattern is then assigned to the given cell and used for the local calibration of the energy scale. It is worth noting that the local energy calibration is associated with the region of the investigated object and not with the specific region of the GEM detector.The calibration coefficients obtained for the given cell are then applied to all 16 pixels of that cell. The energy spectra of all pixels within the cell are then corrected, and the new corrected energy spectrum for the entire investigated area is built. In such a spectrum, an ROI is defined for each distinguished peak. For each pixel, the total number of counts within the given ROI is calculated, and the intensity map is built for this particular ROI corresponding to the distribution of the given element in the investigated object.The above described procedure of peak findings is error prone due to low statistics and a lack of specific energy lines in some cells. The wrong assignment of the peaks will result in wrong energy calibration. The cells with wrong calibration occurred in the ROI maps as sticking out from the surrounding area. Thus, the calibration factors for such pixels can be replaced by the ones derived from the neighbouring pixels. An automated procedure was worked out and implemented to perform such corrections. It may happen that the automatic procedure does not work because of very low statistics in several adjacent pixels or some noisy channels. In such a case, one can still perform manual correction after inspection of the spectra in the suspected cells. Usually, the percentage of such problematic cells is very low, and the manual intervention in the data calibration procedure is not required.

### 2.5. Factor Analysis

The outcome of the data preparation and calibration procedures described above is a 3-D dataset stored in the form of compressed *numpy* arrays (file type: npz) [49] comprising X- and Y-coordinates of all pixels in the investigated object and the energy spectrum associated with each pixel. Factor analysis was applied to such a dataset using two different factorisation methods: PCA and NMF. Although direct access to implementations of these two methods in Python is possible by embedded *scikit-learn* functions [50], we used an indirect approach instead and imported them from the *hyperspy* library [51]. This open-source module also comprises *scikit-learn* solutions, but its advantage is that it reduces significantly the time of all the necessary calculations associated with transformation of datasets before and after the decomposition procedure, such as, for example, changing the dimensions of input and output data matrices. Both factorisation techniques are based on the assumption that the mixture model of individual spectra in a complex spectrum and the spatial maps are linear. Each method provides us with the eigenvectors (basis vectors) and the loadings, which describe how much each data point contributes to a particular component. Interpretation of such results for complex XRF spectra is, however, not straightforward. In particular, the PCA returns basis vectors with positive and negative values, while the XRF spectra comprise only positive values. The NMF method overcomes this limitation by introducing non-negative constraints.

When analysing the spectra measured with the GEM detector, one has to take into account the escape peaks associated with each characteristic energy line. These escape peaks are not always easy to identify, particularly when the energy resolution is limited. On the other hand, they are associated with a given element in the same way as the main characteristic peak, and one may expect that the factorisation analysis will handle this correlation properly.

A critical parameter to be specified for each analysis is the number of basis vectors (components or factors) that should be equal to the number of elements, for which XRF signals are expected to be present in the measured spectra. Usually, an optimum number for a given dataset can be established by observation of the results for different trials and comparing these results with the measured spectra. If the number is too small, some significant components can be lost. On the other hand, introducing a high number may result in assigning particular eigenvectors to some artefacts related to the non-perfect operation of the apparatus. In the case of our detector, the discussed earlier problems with the charging-up effects and energy calibration may also result in some distortion of the measured spectra.

### 2.6. Investigated Objects

To demonstrate the capabilities of our detection system and different data analysis techniques, two historical oil paintings, shown in Figure 7, were investigated. Both objects are part of the collection of the National Museum in Krakow. They were selected to represent two different historical periods and to investigate performance of the spectrometer for different complexities of the pigments used by the artists. The first painting “Portrait of Jan III Sobieski in Karacena Scale Armour” is dated around the year 1700, but its author remains unknown. Its dimensions are 65 cm × 81 cm. The edge of the painting implies a rather simple and limited set of pigments used.

The second object “Portrait of Mieczysław Gąsecki” is a modern Polish painting dated to 1923, and one may expect a larger set of the used pigments. The author of this object is Polish painter Jacek Malczewski. The painting has dimensions of 71 cm × 50 cm. For each painting, only a part of the total area was investigated. The investigated regions are indicated by the white dashed lines, and their zoomed views are shown in Figure 7. The measurement settings for the two paintings are summarised in Table 1.

## 3. Results and Discussion

The measurement data collected for the two paintings were processed and analysed following the procedures described in Section 2.4 and Section 2.5. For each painting, the cumulative spectrum and the results of factor analysis are presented. For each cumulative spectrum, we defined the ROIs and then we built the maps corresponding to each ROI. The factor analysis yields the maps of identified components representing the maps of elemental distributions.

### 3.1. “Portrait of Jan III Sobieski in Karacena Scale Armour”

Figure 8 shows the cumulative XRF spectrum and the results of the NMF and PCA analysis for the painting “Portrait of Jan III Sobieski in Karacena Scale Armour”. Figure 8a shows the total cumulative spectrum collected from the whole investigated area with the selected ROIs. Each ROI corresponds to characteristic energy assigned to the specific element. A particular feature of the GEM detectors that has to be taken into account when analysing the spectrum are the escape peaks. In the case of our detector flushed with the Ar/CO${}_{2}$ gas mixture, the energy of the escape peak is 2.96 keV lower than the main peak for each characteristic energy line. Usually, the escape peak of the lowest measured energy peak is well distinguished from the background. For complex spectra, the escape peaks of higher energy peaks are not always easy to identify. In case of the presented spectrum, the escape peak of the iron line can be easily identified, and an ROI is defined for it as well. In addition, an ROI corresponding to the mercury characteristic line of 10 keV was added, although no clear signature of such a signal is visible in the spectrum. However, both factor analyses indicate the presence of such a signal.

Figure 8b,c show the factors obtained from the NMF and PCA analysis, respectively. Both factorisation analyses reveal signals assigned to the same set of elements: iron, copper, mercury, and lead. It should be noted that the positive values of the NMF components allow one to link directly the dominating peaks of the factor distributions with the characteristic peaks in the cumulative spectrum. The relative intensities of the factors are different compared to what one could expect from the cumulative spectrum. This is not surprising as, by definition, the factorisation techniques do not provide quantitative results.

The intensity maps for all ROIs and the maps of loadings obtained from factor analyses are shown Figure 9. Careful visual inspection of these maps allowed us to make several observations:The ROI maps show identical distributions of lead and mercury, which is not surprising given some overlap of the Hg and Pb${}_{L_{\alpha }}$ ROIs. The Pb${}_{L_{\beta }}$ maps confirmed that lead is indeed present in the painting, but the question about the presence of mercury would remain open if we had only the ROI map. The employment of factor analysis clearly helped us to resolve this particular question. In particular, the NMF loading shows a very well distinguished mercury signal in the lower left-hand side area of the picture. This signal is also visible in the PCA map, but it was negative, and the intensity of its map is inverted with respect to the NMF map.The limitation of the ROI method is also visible in the case of the copper distribution. The map suggested a uniform distribution of copper, which is not expected in the painting, but it is present in a small amount in the form of a grid on the GEM foils. This fake uniform copper distribution can be explained if one notices that the escape peak associated with the Pb${}_{L_{\alpha }}$ line has an energy of 7.59 keV, which is close to the copper line of 8.05 keV. However, based on the ROI analysis alone, we cannot assume *a priori* that the selected copper ROI is dominated by the escape peak of the lead line. The copper grid structure is very clearly visible on the map obtained from the NMF analysis. It is worth noting that the copper grid is visible also on the lead maps as stripes with reduced intensity, which are due to absorption of the lead characteristic radiation in the copper stripes.For the iron distribution, all three techniques gave similar results; however, one can notice that the best selectivity (contrast) is delivered by the NMF factor analysis.

### 3.2. “Portrait of Mieczysław Gąsecki”

Figure 10 shows the cumulative XRF spectrum and the compositions of factors returned by the NMF and PCA factorisation algorithms for the painting “Portrait of Mieczysław Gąsecki”. Figure 10a shows the cumulative spectrum collected from the whole investigated area with the selected ROIs. Because of the limited energy resolution, there are two possible candidates for the highest peak in the middle of the spectrum. We expect some background copper signal from the detector structure; however, the peak position does not match exactly the Cu${}_{K_{\alpha }}$ line. Thus, the peak can be assigned either to copper or to zinc. Therefore, we introduced two ROIs, for copper and for zinc, to be considered for further investigation. Consequently, we also introduced two ROIs corresponding escape peaks of the copper and the zinc lines. They match well with the broad peak in the range of 4.5 to 5.5 keV, although the intensity of this peak is rather low compared to the background. The iron ROI needs an additional comment. In the cumulative spectrum, there is no sign of the iron signal; however, such a signal occurs in the outputs of the factor analyses. Therefore, this ROI is taken into account in further analysis. The escape peak of the Fe line is also hardly distinguished from the background, and the ROIs for this peak was added only in the second iteration, after observation of the factorisation analyses results. Figure 10b,c show the factors obtained from the NMF and PCA analysis, respectively. Both factorisation techniques reveal signals assigned to the same set of elements: iron, copper, zinc, and lead.

The maps for all ROIs and the maps of loadings obtained from factor analyses are shown in Figure 11. It is worth noting a few particular effects observed after careful examination and comparison of the presented maps:

The ROI maps obtained for copper and zinc are practically identical, and the ROI analysis alone does not provide us with any hint regarding which one is true. Based on the ROI analysis, one could also make a hypothesis that a mixture of pigments comprising copper and zinc was used, although such a combination was not expected given our knowledge regarding the techniques used by the author of the painting. Both factorisation analyses NMF and PCA separate the two signals very clearly. The particular pattern of small zinc reach shapes, like the one on the upper-left-hand side corner, is clearly visible on the ROI map as well on the NMF and PCA maps.Both factor analyses indicate a uniform distribution of copper, which is not expected, except the signal from the copper grid included in the GEM foils. However, in the case of NMF maps, one can notice that the copper map is very similar to the lead one. The shape of the factor associated with this map includes the two peaks, which match well with the escape peaks of Pb${}_{L_{\alpha }}$ and Pb${}_{L_{\beta }}$ lines. In this particular case, the almost completely uniform copper PCA map seems to be more correct. On the other hand, the PCA map for lead seems to also include the zinc signal. Thus, the PCA analysis clearly failed to separate these two components.The advantage and usefulness of the factor analyses are very clear in the case of the iron maps. As mentioned before, in the cumulative spectrum, there is no sign of the iron signal, and, based only on the cumulative spectrum, there is no indication to define an ROI in this energy range. However, both NMF and PCA analyses give factors that can be associated with the iron energy line of 6.4 keV. Thus, we extracted the map for this ROI, which indeed confirmed the distribution of iron as obtained from the factor analyses.

## 4. Conclusions

In the presented case studies, we demonstrated that the factor analysis methods NMF and PCA are very effective tools for analysis and interpretation of the XRF images acquired with a full-field imaging spectrometer at limited energy resolution. By combining these two methods with the simple ROI analysis, one can remove ambiguities in the identification of different elements due to the limited energy resolution of the GEM detector used for simultaneous position-sensitive and energy-dispersive detection of X-rays.

Interpretation of the results given by the NMF and the PCA analysis, each one considered separately, is not straightforward and clearly may lead to wrong conclusions in some particular cases. Therefore, one needs to start with the ROI analysis, which allows one to link the shapes of the factors obtained from the factorisation with the characteristic X-ray energy lines. The iterative use of the three methods can enhance significantly the selectivity of the elemental analysis in case of employing an imaging spectrometer with limited energy resolution. It is worth noting that each art object has its specificity and represents a different problem from the point of view of heritage science. As underlined by other studies, e.g., [52], usually a priori knowledge about the investigated object has to be taken into account, and the factorisation methods should be considered as additional supporting tools.

The comparison of the results obtained by the PCA and by the NMF analysis for the object discussed in this article indicates clearly that the NMF is more robust and provides correct factorisation even in the cases with low quality input data. The NMF analysis provides a factor representation, which can be directly linked to the energy spectra, although they cannot be interpreted quantitatively in terms of the concentration of elements in the investigated objects. However, the intensity maps obtained from the ROI and the NMF are qualitatively very similar. The PCA method returns basis vectors including negative values, which cannot be linked directly to the intensity of X-rays given that the spectra delivered by the spectrometer comprise only positive signals. Difficulties with interpretation of the PCA results have been discussed also by other authors [32,53], but no systematic approach was proposed. In our case, the maps produced with this factorisation technique usually showed less contrast; however, in some particular cases, they may be helpful for resolving ambiguities in the results obtained from the ROI and NMF analyses.

## Figures and Tables

**Figure 1 sensors-21-07965-f001:**
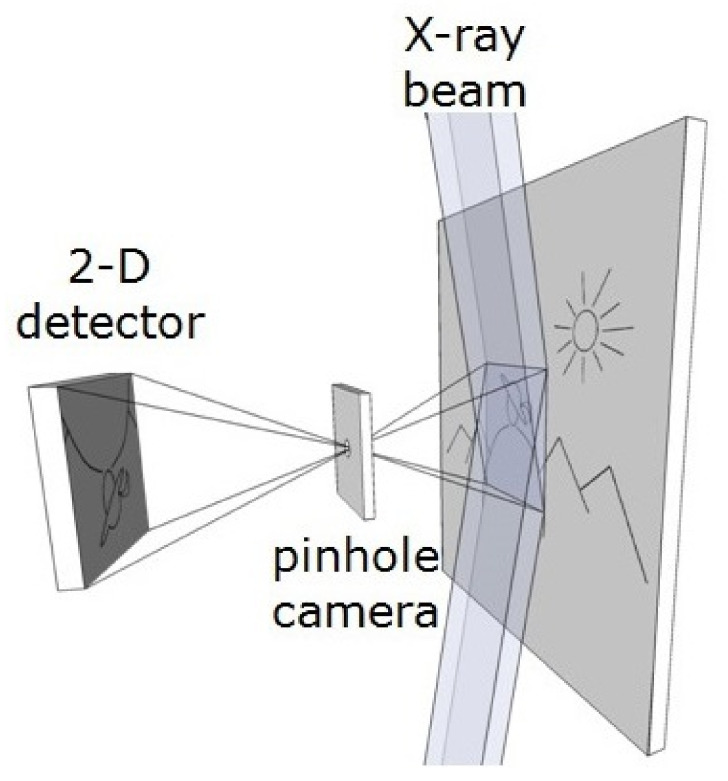
Conceptual view of the full-field XRF imaging spectrometer [31].

**Figure 2 sensors-21-07965-f002:**
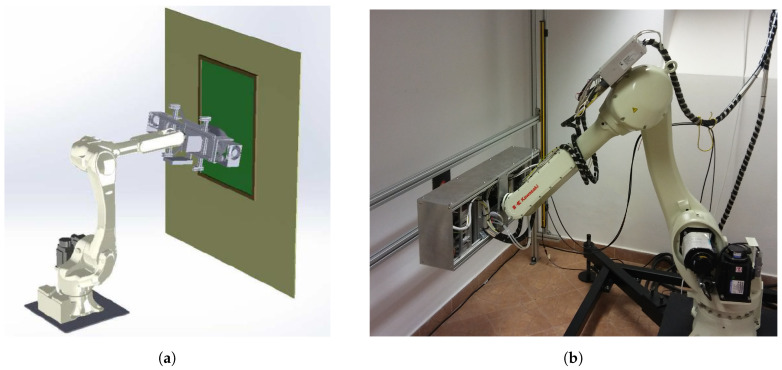
Set-up of the full-field XRF spectrometer. (**a**) Schematic view of the set-up using an industrial robot arm. (**b**) Photo of the spectrometer head mounted on the robot arm.

**Figure 3 sensors-21-07965-f003:**
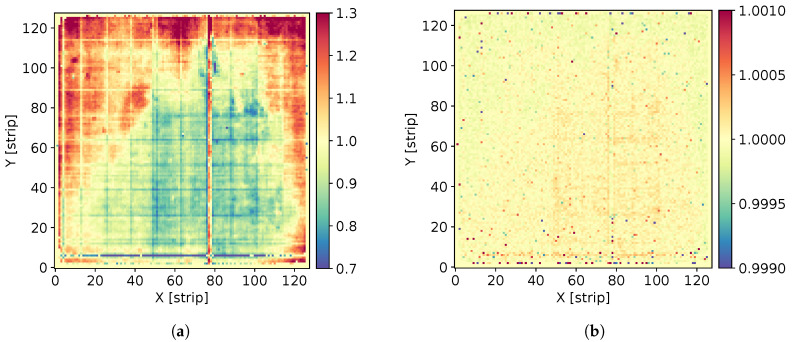
Maps of gas amplification factor across entire detector area: (**a**) extracted from measurements of the characteristic copper radiation of 8.05 keV; (**b**) after off-line correction.

**Figure 4 sensors-21-07965-f004:**
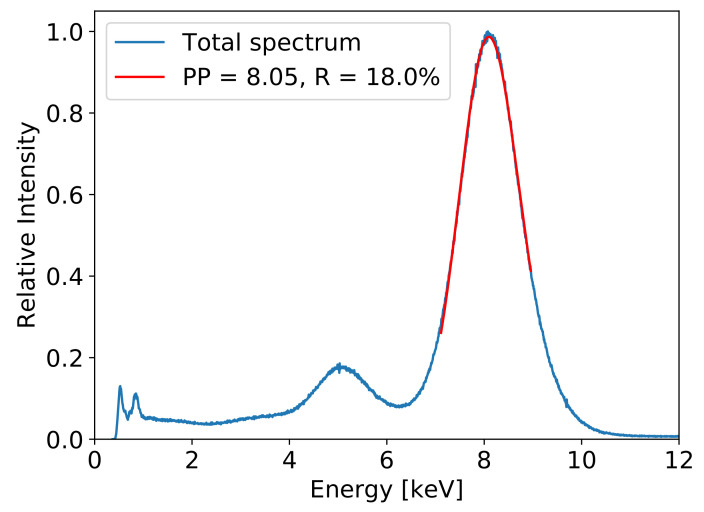
Cumulative spectrum of 8.05 keV X-rays for the GEM detector flushed with Ar/CO${}_{2}$ (75/25) gas mixture and irradiated over the full area of 10 × 10 cm${}^{2}$.

**Figure 5 sensors-21-07965-f005:**
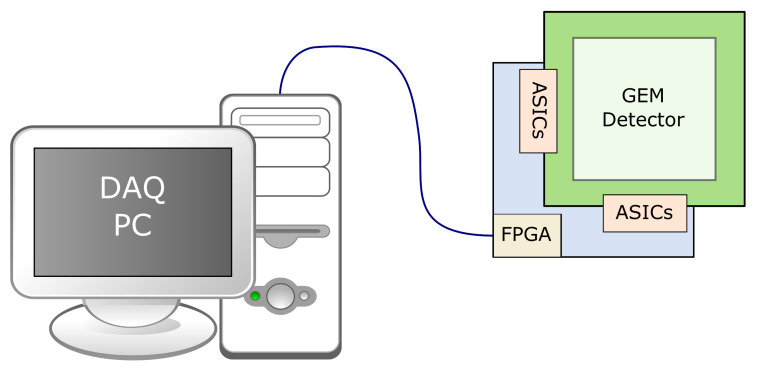
Simplified block diagram of the readout system.

**Figure 6 sensors-21-07965-f006:**
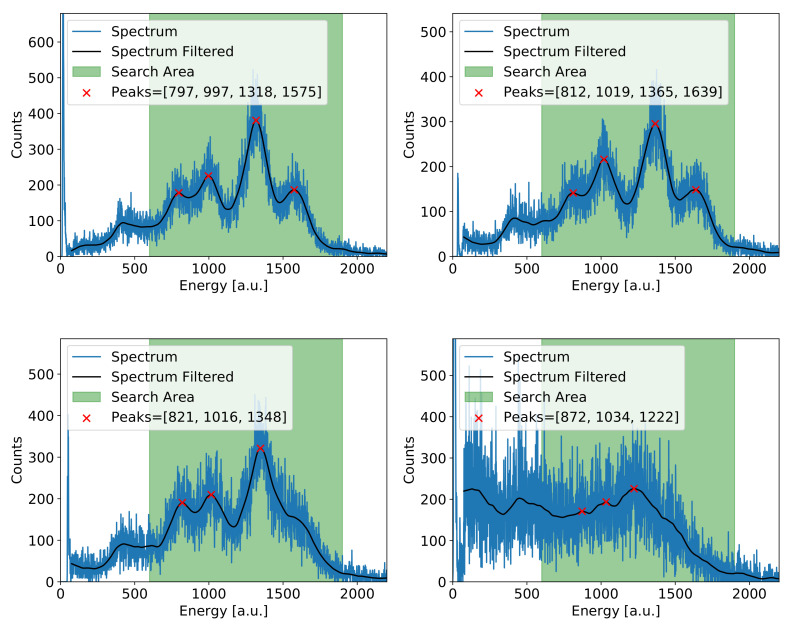
Examples of four different cell spectra with the results of peaks searching.

**Figure 7 sensors-21-07965-f007:**
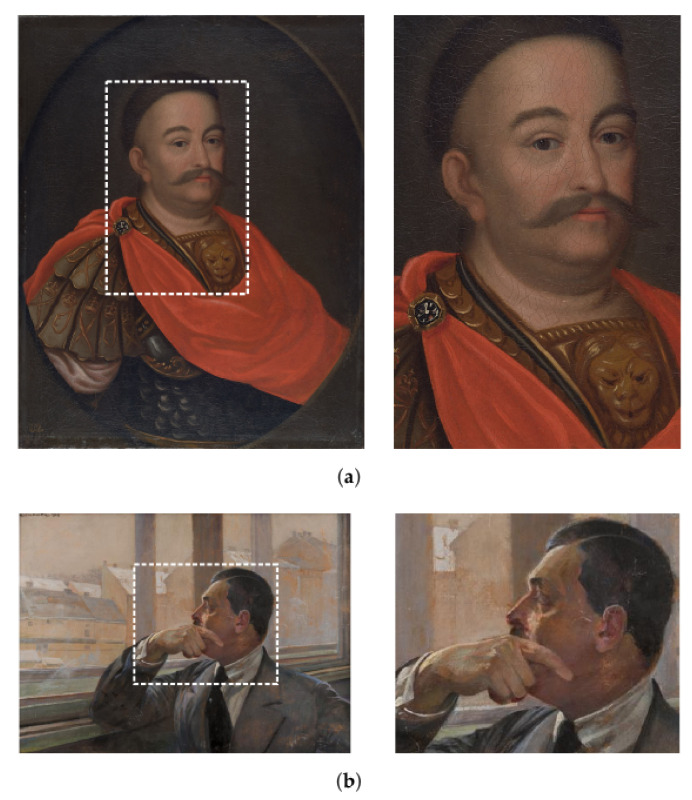
Photographs of the investigated historical paintings. The investigated areas are marked with white line rectangles and shown on the right-hand side pictures. (**a**) “Portrait of Jan III Sobieski in Karacena Scale Armour”. (**b**) “Portrait of Mieczysław Gąsecki”.

**Figure 8 sensors-21-07965-f008:**
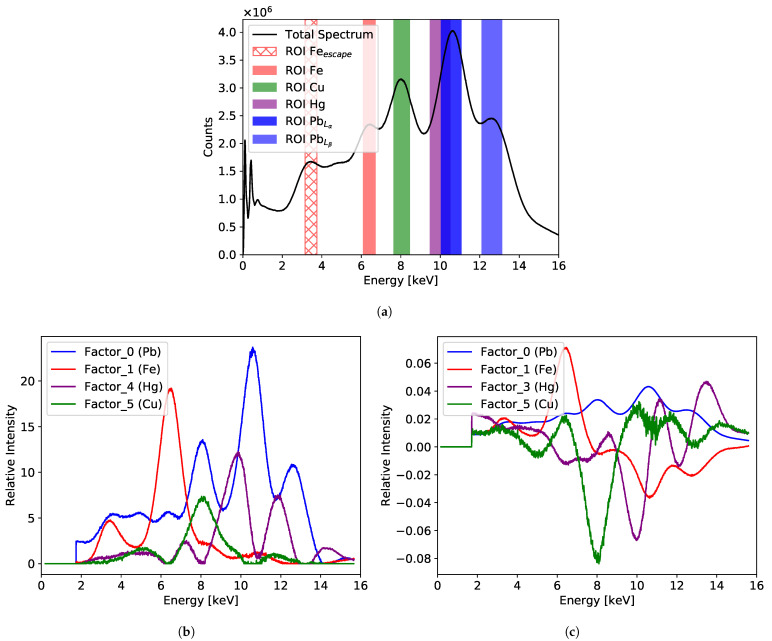
Results for the painting “Portrait of Jan III Sobieski in Karacena Scale Armour”: (**a**) cumulative spectrum for the whole measured area with marked six ROIs; (**b**) factor composition obtained from the NMF analysis; (**c**) factor composition obtained from the PCA analysis.

**Figure 9 sensors-21-07965-f009:**
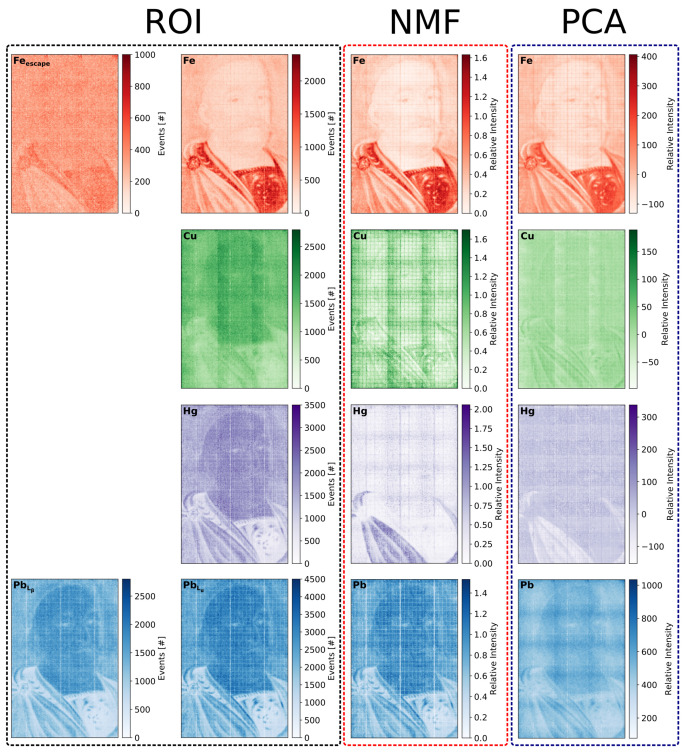
Comparison of the elemental distribution maps obtained for the “Portrait of Jan III Sobieski in Karacena Scale Armour” painting by three different analysis methods: ROI, NMF, and PCA.

**Figure 10 sensors-21-07965-f010:**
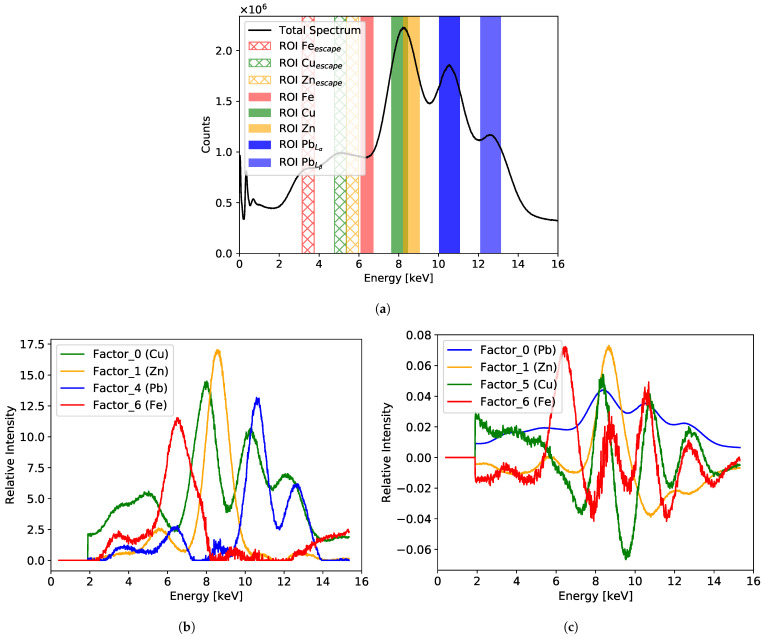
Results for the painting “Portrait of Mieczysław Gąsecki”: (**a**) total cumulative spectrum for the whole measured area with marked eight ROIs, (**b**) factor composition obtained from the NMF analysis, and (**c**) factor composition obtained from the PCA analysis.

**Figure 11 sensors-21-07965-f011:**
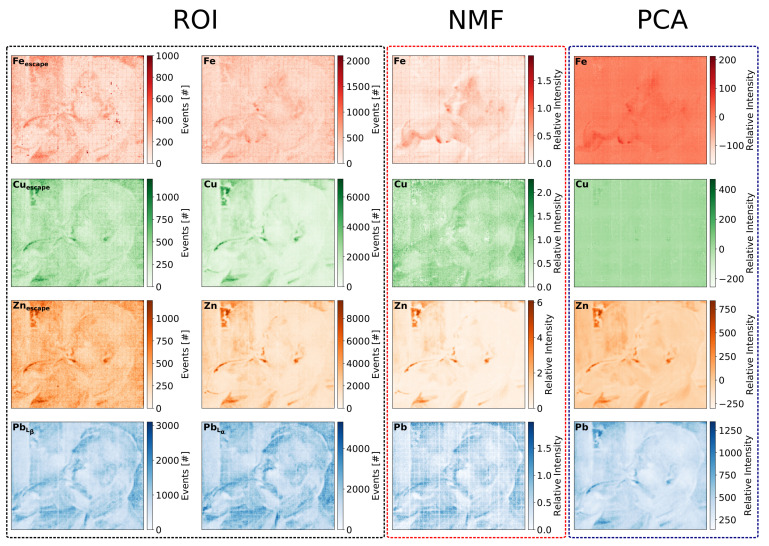
Comparison of the elemental distribution maps obtained for the “Portrait of Mieczysław Gąsecki” painting by three different analysis methods: ROI, NMF, and PCA.

**Table 1 sensors-21-07965-t001:** Measurement settings.

Painting	No. of Frames	Acquisition Time for One Frame (min)	Measured Area (cm${}^{2}$)
“Portrait of Jan III Sobieski in Karacena Scale Armour”	15	20	43 × 26
“Portrait of Mieczysław Gąsecki”	12	20	23 × 29

## Data Availability

Data available on request. The data presented in this study are available on request from the corresponding author.

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
