# Peer review of "Application of Factorisation Methods to Analysis of Elemental Distribution Maps Acquired with a Full-Field XRF Imaging Spectrometer"

_sensors, 2021, doi:10.3390/s21237965_

Round 1

Reviewer 1 Report

This submission investigates potential applications of factor analysis methods, including region of interest (ROI), non-negative matrix factorisation (NMF), and principal component analysis (PCA), for processing X-Ray Fluorescence data. While the study could be of interest for related field, the quality of the paper must be significantly improved before being considered for publication in Sensors:

  1. The English writing should be improved and extensive editing of English language is needed. There are many typos and grammar errors in the manuscript, e.g., in line 1, "application" should be "applications".
  2. The introduction is unbalanced. Little is mentioned about ROI, NMF and PCA. Related works should be reviewed. Especially, since NMF is proven to be more robust and effective in this paper, recent development and applications of NMF methods should be discussed, e.g., "Collaborative Clustering of Subjects and Radiomic Features for Predicting Clinical Outcomes of Rectal Cancer Patients" in IEEE ISBI'19

Author Response

We would like to thank you for the constructive comments. Regarding our answers to your questions and remarks, please see the attachment.

Reviewer 2 Report

Review for the manuscript:

Entitled: "Application of factorisation methods to analysis of elemental distribution maps acquired with a full-field XRF imaging spectrometer"

for Sensors.

With ID: sensors-1439320

Dear Authors,

Thank you for your manuscript.

General comments

Comments for the Authors,

This work is well within the scope of Sensors, and it may be of interest to most of the readers of this journal. The introduction section should be revised, since Turnitin shows a similarity index of 14%. And this value mostly comes from this section in which text overlaps with previous publications of this group. For example: W. Dąbrowski et al 2016 JINST 11 C12025. Authors should clarify the novelty of this work with previously published material. In the current form, the English is good.

For all the above, and the specific comments below, I have opted to recommend a Major revision for the current form of this work.

Specific comments

Abstract

L3: ‘which provides only limited energy resolution’ Please explain.

L4: ‘In the paper’ -> ‘In this paper’.

In the results section of the abstract, authors do not provide any specific results. Please revise.

Introduction

L20-27, 32-50: Most of the introduction section text overlaps with previous publications of this group. For example: W. Dąbrowski et al 2016 JINST 11 C12025. Please revise.

L28: ‘There is a commercially available instrument from Bruker Nano GmbH known as M6-Jetstream [7], which is widely used nowadays [6,8–11].’ Please provide more details regarding the applicability of this instrument.

L31: ‘For nonflat objects, the spatial resolution gets degraded if the investigated surface is not precisely in the focal plane of the microbeam.’ Please provide an appropriate reference to support this statement.

Materials and Methods

L70: ‘Two X-ray tubes with molybdenum anodes’ this was also stated in L66.

L75: ‘The spatial resolution is determined by the pin-hole diameter.’ Please discuss the influence of the 2mm pin hole on spatial resolution.

L82. Section 2.2. By reading this section (up to L121) I think that this text belongs to the introduction or discussion sections, rather than the materials and methods.

L140: ‘However, our observations consistent with other studies [38] indicate that, in addition to this global charging-up effect, there is another level of gas gain variation if the rate of incident radiation varies in time.’ Please revise this sentence.

L298: ‘The NMF method overcomes this limitation by introducing nonnegative constraints, however, the result of such decomposition is still qualitative’ Please provide an appropriate reference to support this statement.

Conclusions

L439: ‘Interpretation of the results given by the NMF and the PCA analysis, each one considered separately, is not straightforward and clearly may lead to wrong conclusions in some particular cases. Therefore one needs to start with the ROI analysis, which allows us to link the shapes of the factors obtained from the factorisation with the characteristic X-ray energy lines.’ So, if someone uses these three methods in conjunction can draw clear conclusions? Do you need an additional method, or could another method provide more accurate results instead? Please discuss more on the limitations of this methods and also future work that will improve the accuracy of the proposed methodology.

Author Response

(The authors gave the same response as above.)

Round 2

Reviewer 1 Report

In general the quality of the paper has been improved.

Reviewer 2 Report

Review for the manuscript:

Entitled: "Application of factorisation methods to analysis of elemental distribution maps acquired with a full-field XRF imaging spectrometer"

for Sensors.

With ID: sensors-1439320.R1

Dear Authors, 

General comments

Comments for the Authors

My previous comments were addressed; thus, I have opted to recommend the acceptance of the manuscript.

Best regards